# The Prevalence and Association of Cognitive Impairment with Sleep Disturbances in Patients with Chronic Liver Disease

**DOI:** 10.3390/brainsci12040444

**Published:** 2022-03-26

**Authors:** Oana-Mihaela Plotogea, Camelia Cristina Diaconu, Gina Gheorghe, Madalina Stan-Ilie, Mircea-Alexandru Badea, Cristina Cijevschi Prelipcean, Gabriel Constantinescu

**Affiliations:** 1Department 5, “Carol Davila” University of Medicine and Pharmacy, 050474 Bucharest, Romania; plotogea.oana@gmail.com (O.-M.P.); gheorghe_gina2000@yahoo.com (G.G.); drmadalina@gmail.com (M.S.-I.); gabrielconstantinescu63@gmail.com (G.C.); 2Department of Gastroenterology, Clinical Emergency Hospital of Bucharest, 105402 Bucharest, Romania; 3Department of Internal Medicine, Clinical Emergency Hospital of Bucharest, 105402 Bucharest, Romania; 4“Sf. Ioan Cel Nou” Emergency County Hospital, 720224 Suceava, Romania; badea.mircea.alexandru@gmail.com; 5Department of Medical Specialties (I), Faculty of Medicine, “Grigore T. Popa” University of Medicine and Pharmacy, 700111 Iasi, Romania; cristinacijevschi@yahoo.com

**Keywords:** chronic liver disease, cognitive impairment, sleep disorders, psychometric testing, actigraphy

## Abstract

Introduction/Aim. The aim of this study was to assess the prevalence of cognitive impairments and their association with sleep patterns in a cohort of patients diagnosed with chronic liver disease (CLD). Material and methods. The present paper is a prospective cohort study, carried out over a period of 12 months, among patients with various stages of CLD. We evaluated the cognitive function through psychometric hepatic encephalopathy score (PHES), while sleep was assessed by actigraphy and two self-reported questionnaires: Epworth Sleepiness Scale (ESS) and Pittsburgh Sleep Quality Index (PSQI). Results. Seventy-four patients with CLD were considered eligible and were enrolled between December 2020–November 2021. The prevalence of minimal hepatic encephalopathy (MHE) in the entire cohort was 41.9%, and the diagnosis was considered for PHES scores ≤ −3. Patients with cirrhosis recorded significantly lower PHES scores compared to patients with other CLDs but without cirrhosis (−3.19 ± 3.89 vs. 0.19 ± 2.92, *p* < 0.05). Patients who exhibited MHE suffered from poor sleep, daytime somnolence, disturbed nighttime sleep, and low overall sleep efficacy. Patients diagnosed with MHE and undergoing treatment with lactulose and/or rifaximin for prevention of overt hepatic encephalopathy (HE) showed better results in terms of sleep parameters compared to patients diagnosed with MHE but without treatment. Conclusions. This research increases awareness regarding the connection between sleep features and MHE in patients with cirrhosis and other CLDs. A deeper insight into the subclinical stages of HE and associated sleep disturbances is warranted in future studies.

## 1. Introduction and Aim

Liver cirrhosis and other chronic liver diseases (CLDs) represent a major cause of morbidity and mortality worldwide, accounting for a high burden of disability in patients and important costs for health care systems. Globally, over 1.5 billion persons have a chronic liver disease (CLD), mostly caused by non-alcoholic fatty liver disease (NAFLD), chronic viral hepatitis, or alcoholic liver disease (ALD) [1,2]. Regarding the management of these conditions, the focus is prophylaxis by means of vaccination and etiological treatment with antivirals, and it is most important to reduce future risk of developing complications of cirrhosis and hepatocarcinoma [3]. In addition to identifying the etiology and assessing the disease severity, it is important to evaluate other conditions that might contribute to a worse prognosis and interfere with these patients’ health-related quality of life [4,5].

The International Society for Hepatic Encephalopathy and Nitrogen Metabolism consensus stated that hepatic encephalopathy (HE) can show two forms: (i) covert HE, which encompasses minimal hepatic encephalopathy (MHE) and grade I from West Haven classification, and (ii) overt HE, which represents the correspondent of grades II, III, and IV according to West Haven criteria [6]. Taking into consideration the underlying disease, HE may be classified in type A (occurs in acute liver failure), type B (occurs in the presence of portosystemic shunts), and type C (encountered in patients with cirrhosis and portosystemic bypass) [6].

Cognitive impairment, comprising attention, motor speed, and affective and executive functioning, are well known and intensively studied in patients with decompensated cirrhosis as part of the HE scenario [7,8]. However, these dysfunctions have a progressive course, being present but less evident in patients with CLDs in pre-cirrhotic stages or compensated cirrhosis [7,9]. Whereas overt HE is a sign of advanced liver insufficiency since it is clinically obvious, covert HE is difficult to be diagnosed in the absence of neurophysiological and psychometric tests [10]. In the daily practice, MHE has a tremendous importance, as it indicates a high susceptibility to progress to overt HE, which seriously impacts health-related quality of life [11].

Patients with CLDs are also susceptible to sleep disorders, translated into daytime sleepiness, difficulty with falling asleep, reduced sleep efficacy, and frequent nighttime awakenings [12]. These symptoms have a clear impact on the quality of life, contributing to a series of other manifestations related to liver insufficiency, such as ascites, HE, jaundice, pruritus, and fatigue [11,12,13].

The relationship between sleep disorders and HE in patients with CLDs is debatable. It has been estimated that approximately half of the patients with cirrhosis report some type of disturbed sleep pattern commonly related to the clinical manifestations of HE [14,15]. However, some observational studies proved that sleep is impaired in cirrhosis because of reasons other than HE, as they were highly prevalent in patients with normal psychometric performance [16].

The aim of our study was to assess the prevalence of subclinical cognitive impairments in patients with CLDs and their association with sleep characteristics in these patients in order to increase the physicians’ awareness regarding other manifestations of MHE, such as sleep disorders. Both cognitive impairment and sleep abnormalities synergically contribute to a poor health-related quality of life.

## 2. Materials and Methods

### 2.1. Subjects

The prospective cohort study was conducted in the Clinical Emergency Hospital of Bucharest, Romania, between December 2020–November 2021.

We initially recruited 91 patients in accordance with the inclusion and exclusion criteria mentioned below.

The inclusion criteria were adult patients (older than 18 years) with CLDs, namely steatosis, chronic hepatitis, and cirrhosis, already diagnosed in our hospital or newly diagnosed. CLD was diagnosed according to clinical, laboratory, and imaging data (abdominal ultrasound with transient elastography/FibroScan evaluation). The study focused solely on type C HE, subclinically manifested (minimal HE).

The exclusion criteria referred to patients with present/past overt HE (clinically manifested as West Haven ≥ II), type A HE, acute hepatitis, or acute liver failure regardless of the cause; significant alcohol consumption (>140 g/week) in the past two weeks, alcohol withdrawal with psychiatric/neurologic manifestations, end-stage renal disease requiring dialysis, present/past sleep medication or sleep pathology, terminal illness of non-hepatic origin, active/past psychiatric or neurologic conditions (e.g., stroke, brain injury etc.), unstable or hemodynamic or cardiovascular status; patients with uncorrected visual disabilities or blindness; and illiterate subjects or night-shift workers.

We collected data, including demographics, etiology, severity (according to Child-Pugh score), ongoing treatment with lactulose and/or rifaximin for HE, diabetes, smoking status, and history of significant alcohol consumption.

### 2.2. Sleep Measurements

Sleep characteristics were assessed using questionnaires and actigraphy. All patients were asked to complete the Pittsburgh Sleep Quality Index (PSQI) [17] and the Epworth Sleepiness Scale (ESS) [18]. Both questionnaires were provided in Romanian language by Mapi Research Trust [19]. Patients answered the questions under the supervision of the investigators either after regular check-ups or on the day of discharge for hospitalized patients.

The PSQI evaluates the sleep quality of the patient during the previous month. The score divides participants into “good” sleepers and “poor” sleepers. The threshold considered suggestive for impaired sleep quality (“poor” sleep) is >5 points. ESS evaluates daytime sleepiness by indicating the probability of falling asleep in 8 ordinary situations. Sums ≥ 11 points for ESS are considered abnormal, suggestive for daytime somnolence [16,20].

Moreover, we evaluated the sleep characteristics of all subjects enrolled, by actigraphic wrist monitoring. The subjects were instructed to wear an Actiwatch device for 7 consecutive days. We used the Actiwatch Spectrum Pro, developed by Philips Healthcare USA, and purchased from LAG MedTech, Kolmar, Sweden. Sleep parameters were recorded and analyzed by automated Philips Actiware software. Reports included data about bedtime hour, get-up hour, time spent in bed, total sleep time, onset latency, sleep efficacy, wake time after sleep onset (WASO), and number of awakenings during night.

### 2.3. Psychometric Testing

The cognitive impairment was evaluated through psychometric hepatic encephalopathy score (PHES). This score includes 5 paper-pencil tests assessing visuomotor coordination and cognitive and psychomotor processing speed. These tests are the following: Number Connection Test A (NCT-A) and Number Connection Test B (NCT-B), Serial Dotting Test (SDT), Digit Symbol Test (DST), and Line Tracing Test (LTT) [21]. NCT-A, NCT-B, and SDT results were calculated in seconds needed for completion, while DST was measured as points (number). LTT was assessed through 2 separate scores. One score was calculated as the time needed (seconds) to complete the test (LTT-t) and the other one (LTT-e) as the errors made (number). Each test was scored according to age and education-adjusted norms calculated for the Romanian population. Psychometric testing with PHES has already been standardized in Romania by Badea et al., in 2016, with a final score ranging between +6 and −18 [22]. The limit for pathological values of the PHES corresponded to –3 points, and the diagnosis of MHE was considered for scores ≤−3 points. Our study cohort completed all 5 tests in a quiet and well-illuminated room within the hospital. Both self-reported sleep questionnaires and PHES forms were completed either after regular check-ups or on the day of discharge for hospitalized patients, after being instructed by one of the two physician investigators and under their supervision.

The PHES forms together with their instructions were provided to us by Dr. Mircea Alexandru Badea and Prof. Dr. Cristina Cijevschi Prelipcean (Iasi, Romania). The copyright of the tests belongs to Prof. Dr. Karin Weissenborn (Hanover Medical School).

### 2.4. Drop-Out Rate

There was an overall drop-out rate of 18.68%. In total, 17 patients who were eligible according to inclusion criteria were ultimately excluded due to incomplete answers of sleep questionnaires (4 patients), not wearing the Actiwatch device for 7 days continuously (6 patients), and not understanding the instructions for the psychometric tests’ completion (7 patients).

### 2.5. Statistical Analysis

First, we collected all data in Microsoft Excel. Second, we used IBM SPSS Statistics for Windows, version 20 (IBM Corp., Armonk, NY, USA) for statistical analysis. Descriptive analysis was performed to express demographic and clinical variables, the prevalence of MHE, sleep quality, daytime sleepiness, and actigraphic sleep characteristics. Means ± standard deviations and ranges or medians and ranges were used for continuous variables. Categorical variables were expressed as frequencies/absolute numbers with percentages. Subgroup differences were tested with chi-square test and ANOVA unifactorial, whichever was relevant. A *p*-value less than 0.05 was considered significant. For multivariate analysis, we used logistic regression with standard method (also known as enter method) by introducing all the independent variables in the equation simultaneously.

### 2.6. Ethics

The study was conducted in accordance with the Declaration of Helsinki, revised in 2008, for medical research involving human subjects [23]. The Research Ethics Committee of the Clinical Emergency Hospital of Bucharest approved the study (no. 3928/11.02.2020). By agreeing to complete the questionnaires and forms, the subjects gave their consent to participate in the study. All subjects provided their approval for using the data by signing a written informed consent before wearing the actigraph.

## 3. Results

### 3.1. Baseline Patients’ Characteristics

Out of the 91 patients initially considered eligible, 74 patients with CLDs were ultimately enrolled and included in the statistical analysis. There were 52 males (70.3%) and 22 females (29.7%), with a mean age of 58.89 ± 9.77 years and a mean educational level of 11.47 ± 2.66 years (Table 1).

For a more detailed analysis, we divided the patients according to transient elastographic evaluation (FibroScan) into two subgroups. Group 1 included patients with CLDs, namely steatosis and/or chronic hepatitis, defined according to FibroScan by no fibrosis = F0, mild fibrosis = F1, and moderate fibrosis = F2–F3. Patients included in Group 1 showed no clinical, paraclinical, or ultrasound indicators of cirrhosis. Group 2 included patients with compensated and decompensated cirrhosis, defined by severe fibrosis = F4, who also presented clinical and paraclinical signs of cirrhosis.

The mean age of patients with cirrhosis was significantly higher compared to the mean age of patients with CLDs but without cirrhosis (62.93 ± 7.76 vs. 53.59 ± 9.70 years, *p* < 0.001). In addition, the mean for schooling years was significantly lower for cirrhotic patients than those included in Group 1 (10.71 ± 2.43 vs. 12.47 ± 2.65 years, *p* = 0.004). We did not find any statistically significant differences (*p* > 0.05) with respect to the two groups’ gender, smoking status, history of significant alcohol consumption, or diabetes.

Regarding the causes that led to CLD, including cirrhosis, overall, we found that the predominant etiology of CLD was related to alcohol consumption. Out of the total number of 74 patients, 25 patients (33.8%) had been diagnosed with alcoholic CLD (9 patients without cirrhosis and 16 with cirrhosis) and 14 patients (18.9%) with combined ethanol and viral causes (2 patients without cirrhosis and 12 with cirrhosis). The differences between groups regarding the etiology proved to be statistically significant (*p* = 0.016), as many alcohol-induced CLDs were more frequently diagnosed as cirrhosis.

### 3.2. Cognitive Assessment by Psychometric Testing

All patients enrolled underwent PHES evaluation comprising the six tests mentioned in the section Material and Methods. MHE, defined as PHES score ≤ −3 points, was encountered in 31 patients (41.9%) out of the entire cohort of 74 patients. Significantly more patients with cirrhosis (24 patients/57.1%, *p* = 0.002) presented MHE, having a mean PHES score of −3.19 ± 3.89 compared to patients with CLD but without cirrhosis (7 patients/21.9%), who recorded a mean PHES score of 0.19 ± 2.92 (Table 2).

When comparing the results within the cirrhosis group in terms of compensated vs. decompensated disease (Table 3), we noticed a statistically significant difference between PHES scores (−1.69 ± 3.49 vs. −4.12 ± 3.89, *p* = 0.048).

Twenty-five (59.5%) patients were undergoing some type of treatment for HE prevention (lactulose and/or rifaximin) as follows: 19 patients (73.1%) with decompensated cirrhosis and 6 (37.5%) with compensated cirrhosis.

Furthermore, based on PHES results, we classified all the patients included in the study in two categories (Table 4): without MHE and with MHE, respectively. We found that diabetes and history of significant alcohol consumption were much more prevalent among patients with MHE (*p* < 0.001). Moreover, we observed that patients with alcohol-induced CLD showed a more important cognitive deterioration, with PHES scores ≤ −3, than those suffering from chronic hepatitis B or C or those with NAFLD (*p* < 0.001). Disease severity was also correlated with a higher prevalence of MHE (*p* = 0.001).

Fifteen patients (48.4%) patients diagnosed with MHE were undergoing prevention therapy with Lactulose and/or Rifaximin for HE. However, most of the patients with MHE had decompensated cirrhosis (17 patients/70.8% out of 31 patients).

### 3.3. Associations between MHE and Sleep Characteristics

We analyzed the relationship between the presence of MHE and sleep patterns as evaluated by means of PSQI, ESS, and actigraphic monitoring (Table 5). Patients with MHE recorded significantly higher scores for PSQI, with mean values of 8.77 ± 3.57 points compared to non-MHE patients, who had an average score of 5.07 ± 2.31 (*p* < 0.001). An important difference was additionally observed for daytime somnolence valuated by ESS questionnaire. In total, 64.5% (20 patients) of the patients with MHE had ESS scores ≥ 11. The mean score for ESS was significantly lower for non-MHE patients (6.42 ± 4.32) compared to MHE patients (11.39 ± 2.97) (*p* < 0.001).

By analyzing the actigraphic parameters recorded, we observed a more delayed bedtime hour for patients with MHE and higher values for WASO without being statistically different any of these variables between the two sub-cohorts (*p* > 0.005). Nevertheless, the get-up time was significantly earlier for non-MHE patients (*p* = 0.003), who also experienced better onset latency and less numbers of awakenings per night and higher overall sleep efficacy (*p* < 0.001).

Furthermore, we investigated the predictors of MHE regarding sleep parameters, which proved to statistically differentiate patients without MHE from those with MHE. Consequently, by multiple regression analysis, we noticed that PSQI, ESS, get-up time, the percentage of sleep efficacy, onset latency, and the number of awakening episodes during night could highly predict (86.80%) the presence of MHE (Table 6, Figure 1).

Taking into consideration the treatment for HE prevention (lactulose and/or rifaximin), we investigated its effects upon different sleep parameters. Hence, we noticed that patients who presented MHE but were under prophylactic treatment recorded lower scores in both PSQI and ESS evaluation (Figure 2A,B) than patients with MHE who did not undergo treatment (7.20 ± 2.88 vs. 10.25 ± 3.60 for PSQI, 10.20 ± 2.27 vs. 12.50 ± 3.18 for ESS).

In terms of actigraphic monitoring (Figure 3A,B), we noticed that sleep efficacy in patients with MHE was higher among patients undergoing prophylactic treatment compared to those who were taking no treatment for HE prevention (77.56 ± 4.17 vs. 75.67 ± 3.69). Moreover, patients with MHE who were under treatment had fewer awakenings per night in contrast with those who were not taking treatment for HE prevention (40.90 ± 9.46 vs. 48.04 ± 6.64).

### 3.4. Sleep Characteristics and Psychometric Testing in Different Subgroups

We performed detailed analyses between various subgroups based on their associated conditions (diabetes and alcohol consumption).

Poor sleep quality and daytime somnolence were significantly more prevalent in patients with diabetes than in non-diabetic patients (*p* < 0.001) (Table 7). Moreover, the actigraphic measurements showed that diabetic patients had lower sleep efficacy, prolonged onset latency, and significantly more awakenings episodes during night (*p* < 0.001). Regarding MHE, PHES scores were significantly higher in non-diabetic patients compared to diabetics (−0.21 ± 2.99 vs. −5.32 ± 3.31, *p* < 0.001).

Similarly, we investigated sleep characteristics and psychometric results of patients with history of significant alcohol consumption by comparing them with patients who were not alcohol drinkers (Table 8). Thus, those who consumed alcohol had significantly higher scores for both PSQI (*p* = 0.015) and ESS (*p* = 0.009). Regarding the actigraphy evaluation, only two parameters proved to have statistical significance: bedtime hour and total sleep time. Bedtime hour was more delayed for alcohol consumers (*p* = 0.002), while the total sleep time was significantly shorter compared to non-drinkers (*p* = 0.04). MHE was more frequent among alcohol consumers, who also recorded lower PHES scores compared to non-consumers (−5.24 ± 3.39 vs. −0.34 ± 3.10, *p* < 0.001).

## 4. Discussion

This study represents an ongoing prospective analysis regarding the prevalence of MHE among patients diagnosed with CLDs (steatosis, hepatitis, and cirrhosis). In addition, the study describes patients’ sleep characteristics by wrist actigraphic assessment and self-reported validated questionnaires (PSQI and ESS) in relation with patients’ cognitive performance according to the PHES used to diagnose MHE.

Liver stiffness in CLDs is mainly characterized through non-invasive methods, with transient elastography being one of the most preferred tools in clinical practice. According to BAVENO VII Consensus, the recently introduced term of compensated advanced chronic liver disease (cACLD) is meant to describe the ongoing severe fibrosis by means of transient elastography. Therefore, it is possible to early detect patients with cACLD, who are at risk of decompensation and clinically significant portal hypertension [24].

Portosystemic encephalopathy syndrome test, also referred to as the PHES, was first developed and standardized in Germany [25]. PHES is widely considered the gold-standard method in assessing MHE. This has a 96% sensitivity for diagnosing HE and a specificity of 100% when comparing patients with clinically overt HE with healthy controls [25]. The PHES is easy to complete in 10–20 min because it does not require trained personnel or advanced equipment. Moreover, it has already been standardized in various countries, such as Romania [22], Spain [26], Turkey [27], Italy [28], USA [29], France [30], etc.

The prevalence of MHE using PHES has been intensively investigated among patients with cirrhosis and is largely varying among different populations, from 25% up to 80% of the patients [25,26,27,28,29,30,31,32]. On the contrary, studies on MHE in patients with early stages of CLDs or pre-cirrhotic stages are scarce. Among the subjects enrolled in our study, with different stages of CLD from steatosis to decompensated cirrhosis, the prevalence of MHE was 41.9%. More specific, the prevalence of MHE in the cirrhosis cohort was 57.1%.

Recently, researchers from Spain reported that 32% of patients with NAFLD show cognitive impairment at completing psychometric tests. In their study, the authors found that comorbidities, such as diabetes or metabolic syndrome, enhance the prevalence of MHE [9]. This finding indicates that MHE may be attributable to additional factors apart from liver disease because neuropsychological or neurophysiological alterations are not pathognomonic for HE. Several associated conditions (e.g., diabetes mellitus, renal failure, hyponatremia, sepsis, Wernicke’s encephalopathy) contribute to worsening the cognitive impairment in patients with HE [33]. The present study analyzed additional variables, such as diabetes, smoking status, and alcohol consumption. Our results showed an association between the presence of diabetes, history of alcohol consumption, and MHE. Moreover, the etiology related to alcohol was the most frequently encountered in patients with CLDs and MHE.

Regarding sleep disorders, lately, there is increasing evidence about their relationship with MHE. The concept of “sleep-wake inversion”, translated through restless nights and daytime somnolence, has been considered a manifestation of overt hepatic encephalopathy [34]. There are numerous studies that describe the presence of sleep disorders among patients with CLDs aside from cirrhosis and regardless of overt HE [35,36]. However, studies that describe sleep-wake disturbances in patients with CLDs in relationship with the presence MHE are scarce. In a study conducted in India, authors showed that scores recorded by ESS were significantly higher in patients diagnosed with MHE through means of PHES score compared to patients without MHE. Moreover, sleep quality measured by PSQI score was worse in MHE patients in comparison to non-MHE ones [37]. In another study including patients with MHE, researchers found strong correlations between many subjective aspects of sleep quality and objective polysomnographic data [38]. Their results confirmed that MHE patients suffer from impaired quality of sleep, prolonged time needed to fall asleep, and low sleep efficacy. Additionally, patients showed daytime functional abnormalities, indicating that MHE patients suffer from multiple subjective dyssomnias [38]. Similarly, our findings showed better scores for PSQI and ESS and improved actigraphic parameters for patients that did not exhibit MHE in comparison to MHE patients. Moreover, we observed that ongoing treatment for HE prevention had a clear benefit regarding sleep in patients diagnosed with MHE.

This study resides several limitations. In the first place, the study cohort is made only of patients with a diagnosis of CLD. There is no control group. Yet, the study’s aim was to report the prevalence in subjects with assumptive dysfunctions. Second, as we enrolled patients from an emergency hospital, a great number was represented by patients with decompensated liver cirrhosis. Third, we did not track the reasons for decompensation, biochemical parameters, or the medication, which might have significantly influenced the results. For example, we did not specifically evaluate which treatment for HE prevention influenced the sleep parameters or other variables analyzed within the study. Fourth, there are some limitations owed to questionnaires and actigraphy. Because these are subjective and semi-objective methods, they might create bias by being overestimated. Ultimately, we did not create specific subgroups according to the non-invasive evaluation of liver stiffness, which definitely would have brought valuable information for diagnosis of cACLD and clinically significant portal hypertension.

The present study assessed by means of PHES the cognitive impairments of patients with chronic liver disease and cirrhosis and their associations with sleep abnormalities. This research demonstrated that Romanian patients with CLDs and MHE suffer from poor sleep, daytime somnolence, disturbed nighttime sleep, and low overall sleep efficacy. Patients with MHE who were undergoing prophylactic treatment with lactulose and/or rifaximin had better PSQI and ESS scores, higher sleep efficacy, and fewer episodes of awakening during the night. Therefore, treatment should be taken into consideration in patients with MHE not only to prevent progression to overt HE but also to improve sleep parameters.

Future studies are warranted to investigate which specific treatment improves sleep and cognitive functions and prevents MHE progression to clinical HE. Moreover, it is mandatory to analyze other contributing factors and provide greater insight into treatment strategies that could make a difference in ameliorating sleep- and health-related quality of life among patients with CLDs.

## 5. Conclusions

The impact of CLDs must be broadly appreciated by taking into consideration cognitive impairment in subclinical stages and associated sleep disturbances, which definitely worsen the patients’ quality of life, daily function, and prognosis. Therefore, this study increases awareness amongst clinicians in order to understand the importance of the “unseen” problems of CLDs. There is an indisputable need for further research in this area by investigating the factors contributing to sleep abnormalities and cognitive disorders in different stages of chronic liver diseases.

## Figures and Tables

**Figure 1 brainsci-12-00444-f001:**
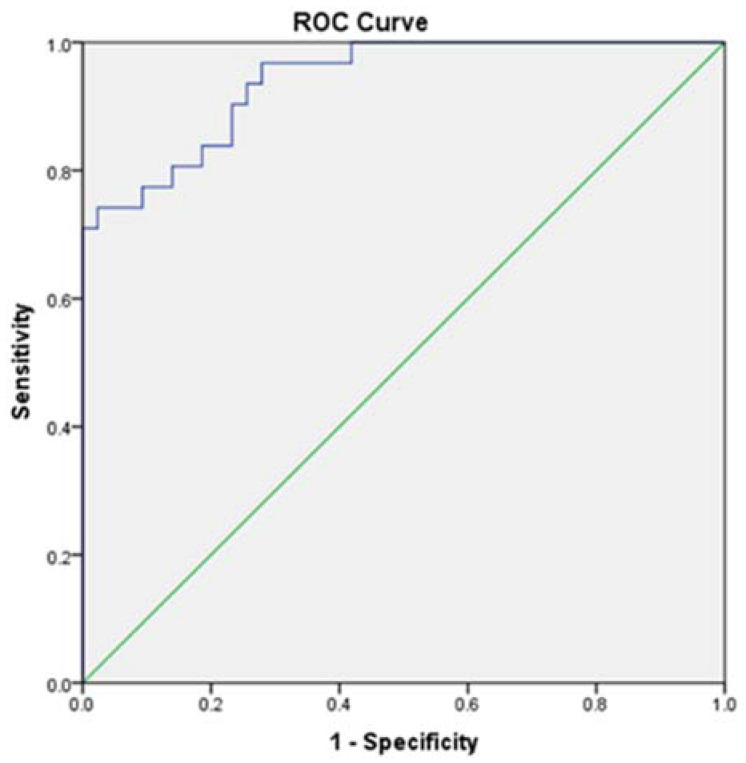
ROC Curve for sleep predictors of MHE (AUC = 0.940; sensitivity = 0.839; specificity = 0.814).

**Figure 2 brainsci-12-00444-f002:**
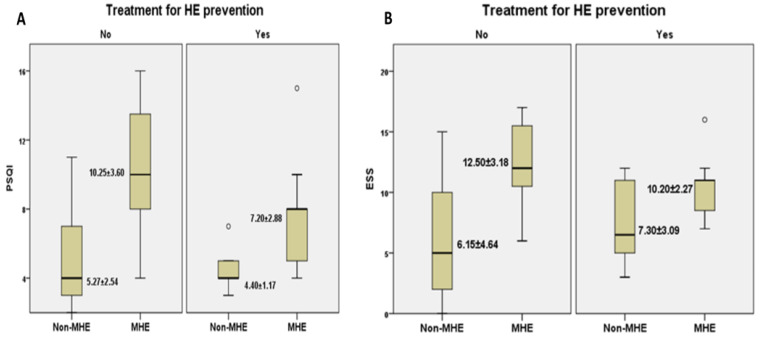
The relationship between MHE and self-reported questionnaires’ mean score: PSQI (**A**) and ESS (**B**) depending on the presence/absence of preventive treatment for HE. Legend. PSQI, Pittsburgh Sleep Quality Index; ESS, Epworth Sleepiness Scale; HE, hepatic encephalopathy.

**Figure 3 brainsci-12-00444-f003:**
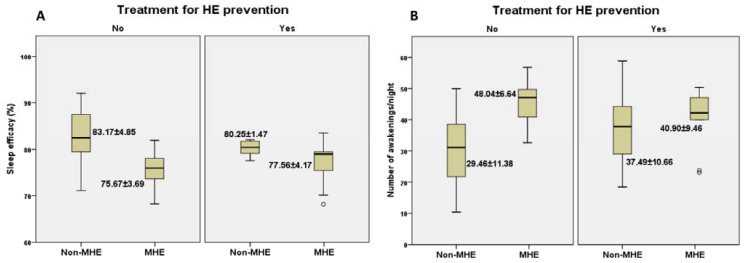
The relationship between the presence/absence of MHE with/without preventive treatment and actigraphic parameters: sleep efficacy (**A**) and number of awakenings/night (**B**).

**Table 1 brainsci-12-00444-t001:** Background characteristics of patients with CLDs enrolled in the study.

	Total (74 Patients)	Group 1 (32 Patients)	Group 2 (42 Patients)	*p* *
Age (mean ± SD)	58.89 ± 9.77	53.59 ± 9.70	62.93 ± 7.76	<0.001
Gender (M/F), *n* (%)	52/22 (70.3/29.7%)	23/9 (71.9/28.1%)	29/13 (69/31%)	0.499
Education (years), mean ± SD	11.47 ± 2.66	12.47 ± 2.65	10.71 ± 2.43	0.004
Alcohol, *n* (%)	21 (28.4%)	7 (21.9%)	14 (33.3%)	0.206
Smoking, *n* (%)	28 (37.8%)	14 (43.8%)	12 (33.3%)	0.25
Diabetes, *n* (%)	22 (29.7%)	8 (25%)	14 (33.3%)	0.303
Etiology, *n* (%)				
Alcoholic	25 (33.8%)	9 (28.1%)	16 (38.1%)	0.016
Viral hepatitis	22 (29.7%)	11 (34.4%)	11 (26.2%)	
Alcoholic + Viral Hepatitis	14 (18.9%)	2 (6.2%)	12 (28.6%)	
NAFLD	11 (14.9%)	8 (25%)	3 (7.1%)	
Autoimmune	2 (2.7%)	2 (6.2%)	0 (0%)	
Disease severity, *n* (%)				
Child A	-	-	16 (38.1%)	-
Child B	-	-	11 (26.2%)	
Child C	-	-	15 (35.7%)	

Legend. NAFLD, Non-Alcoholic Fatty Liver Disease; SD, standard deviation; * *p*: ANOVA and chi-square tests were used for comparison between Group 1 (patients with CLD, without cirrhosis) and Group 2 (cirrhotic patients).

**Table 2 brainsci-12-00444-t002:** Comparison between PHES results of early-stages versus advanced-stages of CLDs.

	Total (74 Patients)	Group 1 (32 Patients)	Group 2 (42 Patients)	*p* *
DST, mean ± SD	34.39 ± 10.15	42.09 ± 7.75	28.52 ± 7.53 *	<0.001
NCT-A, mean ± SD	52.07 ± 17.92	43.28 ± 14.43	58.76 ± 17.56	<0.001
NCT-B, mean ± SD	127.96 ± 35.89	101.56 ± 12.38	148.07 ± 34.92	<0.001
SDT, mean ± SD	69.46 ± 14.59	60.25 ± 16.13	76.48 ± 8.11	<0.001
LTT-t, mean ± SD	99.27 ± 19.19	90.06 ± 18.40	106.29 ± 16.82	<0.001
LTT-e, mean ± SD	37.05 ± 23.60	23.13 ± 17.24	47.67 ± 22.36	<0.001
PHES, mean ± SD	−1.73 ± 3.86	0.19 ± 2.92	−3.19 ± 3.89	<0.001
MHE, *n* (%)	31 (41.9%)	7 (21.9%)	24 (57.1%)	0.002
Treatment for HE, *n* (%)	25 (33.8%)	0 (0%)	25 (59.5%)	<0.001

Legend. DST, Digit Symbol Test (number); NCT-A and NCT-B, Number Connection Tests A and B (seconds); SDT, Serial Dotting Test (seconds); LTT-t, Line-Tracing Test—time (seconds); LTT-e, Line-Tracing Test—errors (number); PHES, psychometric hepatic encephalopathy score; HE, hepatic encephalopathy; SD, standard deviation; * *p*: ANOVA and chi-square tests were used for comparisons between Group 1 (patients with CLD, without cirrhosis) and Group 2 (cirrhotic patients).

**Table 3 brainsci-12-00444-t003:** Comparison between PHES results of compensated with decompensated cirrhotic patients.

	Cirrhosis			
	Total (42 Patients)	Compensated (16 Patients)	Decompensated (26 Patients)	*p* *
DST, mean ± SD	28.52 ± 7.53 *	32.63 ± 7.09	26.00 ± 6.74	0.004
NCT-A, mean ± SD	58.76 ± 17.56	49.94 ± 14.62	64.19 ± 17.23	0.009
NCT-B, mean ± SD	148.07 ± 34.92	133.31 ± 31.11	157.15 ± 34.44	0.03
SDT, mean ± SD	76.48 ± 8.11	73.19 ± 7.33	78.50 ± 8.02	0.038
LTT-t, mean ± SD	106.29 ± 16.82	99.44 ± 17.10	110.50 ± 15.50	0.037
LTT-e, mean ± SD	47.67 ± 22.36	40.31 ± 19.76	52.19 ± 23.01	0.095
PHES, mean ± SD	−3.19 ± 3.89	−1.69 ± 3.49	−4.12 ± 3.89	0.048
MHE, *n* (%)	24 (57.1%)	7 (43.8%)	17 (65.4%)	0.210
Treatment for HE, *n* (%)	25 (59.5%)	6 (37.5%)	19 (73.1%)	0.029

Legend. DST, Digit Symbol Test (number); NCT-A and NCT-B, Number Connection Tests A and B (seconds); SDT, Serial Dotting Test (seconds); LTT-t, Line-Tracing Test—time (seconds); LTT-e, Line-Tracing Test—errors (number); PHES, psychometric hepatic encephalopathy score; HE, hepatic encephalopathy; SD, standard deviation; * *p*: ANOVA and chi-square tests were used for comparisons between compensated and decompensated cirrhotic patients.

**Table 4 brainsci-12-00444-t004:** Clinical data of patients with MHE in comparison with those without MHE.

	Total (74 Patients)	Non-MHE (43 Patients)	MHE (31 Patients)	*p* *
Groups, no. (%)				
Group 1	32 (43.2%)	25 (58.1%)	7 (22.6%)	0.004
Group 2	42 (56.8%)	18 (41.9%)	24 (77.4%)	
Compensated	16 (38.1%)	9 (50%)	7 (29.2%)	0.146
Decompensated	26 (61.9%)	9 (50%)	17 (70.8%)	
Diabetes, no. (%)	22 (29.7%)	3 (7%)	19 (61.3%)	<0.001
Alcohol, no. (%)	21 (28.4%)	4 (9.3%)	17 (54.8%)	<0.001
Smoking, no. (%)	28 (37.8%)	17 (39.5%)	11 (35.3%)	0.81
Etiology, no. (%)				
Alcoholic	25 (33.8%)	6 (14%)	19 (61.3%)	<0.001
Viral hepatitis	22 (29.7%)	19 (44.2%)	3 (9.7%)	
Alcoholic + Viral hepatitis	14 (18.9%)	6 (14%)	8 (25.8%)	
NAFLD	11 (14.9%)	10 (23.3%)	1 (3.2%)	
Autoimmune	2 (2.7%)	2 (4.7%)	0 (0%)	
Disease severity, no. (%)				
Child A	16 (38.1%)	9 (50%)	7 (29.2%)	0.001
Child B	11 (26.2%)	8 (44.4%)	3 (12.5%)	
Child C	15 (35.7%)	1 (5.6%)	14 (58.3%)	
Treatment for HE, no. (%)	25 (33.8%)	10 (23.3%)	15 (48.4%)	0.028

Legend. MHE, minimal hepatic encephalopathy; HE, hepatic encephalopathy; NAFLD, non-alcoholic fatty liver disease; * *p*: chi-square test was used for comparisons between non-MHE and MHE patients.

**Table 5 brainsci-12-00444-t005:** Sleep evaluation of patients with MHE in comparison with those without MHE.

Sleep Characteristics	Total (74 Patients)	Non-MHE (43 Patients)	MHE (31 Patients)	*p* *
PSQI (mean ± SD)	6.62 ± 3.42	5.07 ± 2.31	8.77 ± 3.57	<0.001
Good sleepers (≤5), no. (%)	36 (48.6%)	30 (69.8%)	8 (25.8%)	<0.001
Poor sleepers (>5), no. (%)	38 (51.4%)	13 (30.2%)	23 (74.2%)	
ESS (mean ± SD)	8.50 ± 4.52	6.42 ± 4.32	11.39 ± 2.97	<0.001
<11, no. (%)	43 (58.1%)	32 (74.4%)	11 (35.5%)	0.002
≥11, no. (%)	31 (41.9%)	11 (25.6%)	20 (64.5%)	
Bedtime (h:min ± SD)	22:26 ± 0:45	22:18 ± 0:42	22:38 ± 0:47	0.057
Get-up time (h:min ± SD)	7:49 ± 0:52	7:34 ± 0:47	8:10 ± 0:52	0.003
Time in bed (h:min ± SD)	09:22 ± 0:53	9:16 ± 0:54	9:32 ± 0:50	0.206
Total sleep time (h:min ± SD)	7:40 ± 0:39	7:45 ± 0:37	7:33 ± 0:41	0.193
Sleep efficacy (% ± SD)	80.04 ± 5.14	82.49 ± 4.46	76.64 ± 4.01	<0.001
Onset latency (min ± SD)	21.18 ± 8.40	17.35 ± 7.94	26.48 ± 5.79	<0.001
WASO (min)	38.38 ± 8.46	36.75 ± 8.56	40.63 ± 7.91	0.051
Number of awakenings per night (mean ± SD)	36.51 ± 12.34	32.12 ± 11.91	42.60 ± 10.30	<0.001

Legend. MHE, minimal hepatic encephalopathy; PSQI, Pittsburgh Sleep Quality Index; ESS, Epworth Sleepiness Scale; SD, standard deviation; h, hours; min, minutes; WASO, wake after sleep onset; * *p*: ANOVA and chi-square tests were used for comparisons between non-MHE and MHE patients.

**Table 6 brainsci-12-00444-t006:** Logistic regression analysis for sleep predictors of MHE.

	Multiple Regression			
Variables	OR [95% CI]	Coefficient Beta	*p **	Predicted Percentage
PSQI	1.434 [1.306–1.569]	0.136	0.045	86.80%
ESS	1.247 [1.193–1.361]	0.22	0.032	
Get-up time	1 [1–1.001]	0.001	0.026	
Sleep efficacy	0.803 [0.711–0.904]	−0.220	0.001	
Onset latency	1.212 [1.063–1.383]	0.192	0.004	
Number of awakenings per night	0.944 [0.864–1.031]	−0.138	0.007	

Legend. * *p:* multivariate analysis of variance used for sleep predictors of MHE.

**Table 7 brainsci-12-00444-t007:** Sleep evaluation and psychometric testing of patients with diabetes in comparison with those without diabetes.

Sleep Characteristics and Psychometric Testing	Diabetes (22 Patients)	Non-Diabetes (52 Patients)	*p* *
PSQI (mean ± SD)	9.77 ± 3.22	5.29 ± 2.53	<0.001
Good sleepers (≤5), no. (%)	20 (90.9%)	16 (30.8%)	<0.001
Poor sleepers (>5), no. (%)	2 (9.1%)	36 (69.2%)	
ESS (mean ± SD)	12.00 ± 3.25	7.02 ± 4.18	<0.001
<11, no. (%)	6 (27.3%)	37 (71.2%)	0.001
≥11, no. (%)	16 (72.7%)	16 (28.8%)	
Bedtime (h:min ± SD)	22:26 ± 0:51	22:27 ± 0:042	0.967
Get-up time (h:min ± SD)	7:56 ± 0:54	7:46 ± 0:51	0.465
Time in bed (h:min ± SD)	9:30 ± 0:53	9:19 ± 0:53	0.452
Total sleep time (h:min ± SD)	7:32 ± 0:47	7:44 ± 0:35	0.229
Onset latency (min ± SD)	26.39 ± 7.42	18.97 ± 7.86	<0.001
Sleep efficacy (% ± SD)	76.61 ± 4.72	81.49 ± 4.64	<0.001
WASO (min)	41.33 ± 7.49	37.13 ± 8.61	0.051
Number of awakenings per night (mean ± SD)	44.17 ± 7.99	33.27 ± 12.48	<0.001
PHES, mean ± SD	−5.32 ± 3.31	−0.21 ± 2.99	<0.001
MHE (no., %)	19 (86.4%)	12 (23.1%)	<0.001

Legend. PSQI, Pittsburgh Sleep Quality Index; ESS, Epworth Sleepiness Scale; SD, standard deviation; h, hours; min, minutes; WASO, wake after sleep onset; PHES, psychometric hepatic encephalopathy score; MHE, minimal hepatic encephalopathy; * *p*: ANOVA and chi-square tests were used for comparisons between diabetic and non-diabetic patients.

**Table 8 brainsci-12-00444-t008:** Sleep evaluation and psychometric testing of patients with alcohol consumption in comparison with those without alcohol consumption.

Sleep Characteristics and Psychometric Testing	Alcohol Consumers (21 Patients)	Non-Alcohol Consumers (53 Patients)	*p* *
PSQI (mean ± SD)	8.14 ± 4.23	6.02 ± 2.87	0.015
Good sleepers (≤5), no. (%)	12 (57.1%)	24 (45.3%)	0.442
Poor sleepers (>5), no. (%)	9 (42.9%)	29 (54.7%)	
ESS (mean ± SD)	10.67 ± 3.73	7.64 ± 4.55	0.009
<11, no. (%)	9 (42.9%)	34 (64.2%)	0.120
≥11, no. (%)	12 (57.1%)	19 (35.8%)	
Bedtime (h:min ± SD)	22:52 ± 0:43	22:16 ± 0:42	0.002
Get-up time (h:min ± SD)	8:02 ± 0:53	7:45 ± 0:51	0.208
Time in bed (h:min ± SD)	9:09 ± 0:48	9:28 ± 0:54	0.188
Total sleep time (h:min ± SD)	7:25 ± 0:41	7:46 ± 0:37	0.04
Onset latency (min ± SD)	23.46 ± 8.98	20.27 ± 8.07	0.142
Sleep efficacy (% ± SD)	78.84 ± 5.85	80.52 ± 4.81	0.207
WASO (min)	38.90 ± 7.36	38.17 ± 8.92	0.742
Number of awakenings per night (mean ± SD)	38.23 ± 11.82	35.83 ± 12.58	0.454
PHES, mean ± SD	−5.24 ± 3.39	−0.34 ± 3.10	<0.001
MHE (no., %)	17 (81%)	14 (26.4%)	<0.001

Legend. PSQI, Pittsburgh Sleep Quality Index; ESS, Epworth Sleepiness Scale; SD, standard deviation; h, hours; min, minutes; WASO, wake after sleep onset; PHES, psychometric hepatic encephalopathy score; MHE, minimal hepatic encephalopathy; * *p*: ANOVA and chi-square tests were used for comparisons between alcohol consumers and non-alcohol consumers.

## Data Availability

Data available upon request from the first author.

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
