# Peer review of "The Prevalence and Association of Cognitive Impairment with Sleep Disturbances in Patients with Chronic Liver Disease"

_brainsci, 2022, doi:10.3390/brainsci12040444_

Round 1
Reviewer 1 Report
The authors presented a well written manuscript of a study relevant for the needs of the field. Sleep disturbances can go beyond affecting the quality of life of a patient as they could also reflect cerebral abnormalities such as that of the newly discovered glymphatic system, rendering a more in depth investigation.
I only have some comments:
1. Please also use p values when referring to statistical tests performed on data in the text, easier for the reader.
2. Tables: specify from which comparison the p value is coming from and test used
3. As discussed there are lot of active drinkers and patients with co-morbidities such as diabetes included. Now the definition of MHE is debated as brain dysfunction and MHE can be different. The authors group all patients and divide them between non-MHE and MHE. It is important if possible to show and discuss how sleep quality was affected in all the subgroups e.g. diabetics, alcohol consumers, decompensated etc. to allow for a more clear picture of the cause.
3. Sleep parameters: what about comparison between non treated non-MHE vs treated MHE, since treated non-MHE patients had a worse outcome due to possibly more severe disease stage.
4. Any biochemical parameters correlating with these results? ammonia, inflammation/infection, hyponatremia?
Author Response
"Please see the attachment."

Reviewer 2 Report
Minimal hepatic encephalopathy (MHE) is the earliest form of a spectrum of cognitive abnormalities associated with liver disease and can affect up to 80% of patients who are involved in research (30622374). This undoubtedly indicates the urgency to further explore the topic.
The authors investigating the relationship between cognitive dysfunctions and sleep problems in patients with chronic liver disease (mostly in patients who are found to have MHE) argue that: «Regarding sleep disorders, there are limited data about their relationship with MHE».
It is difficult to agree with this statement since there is good evidence reported as early as 1954 that nocturnal ramblings and sleep rhythm disturbances have been found in patients, regardless of various etiologies and stages of the disease, and a lot of modern studies have confirmed this connection (9462628, 18810597, 29863286, 28070704, 32415046, 23744627, 34276152, 33214928).
At the same time, the authors stated that «treatment with lactulose and/or rifaximin for prevention of clinical HE in patients who exhibit low PHES scores improves sleep quality». These data also do not seem new, since the positive effect of these drugs is not only confirmed in scientific laboratories, but also used in clinical settings to improve disturbances of health-related quality of life and, in particular, sleep disturbances in cirrhotic patients with MHE. (28070704, 9397979, 21646910, 20335583, 21157444, 23877348).
It is worthy of note that the study conducted by the authors and described in the article published in 2021, addressed the same issue and provided similar conclusions: It is stated in the previously published paper that «This is the first study in Romania that assesses sleep by actigraphy in a cohort of patients with different stages of CLD.» and an article submitted to the journal for publication includes the following sentence: «Line 337-339, this is the first study in Romania to assess the connection between sleep features measured through a semi-objective method, namely actigraphy and MHE, evaluated by means of PHES score, in a population of patients with CLDs.
In view of this, the question arises as to what the first study is: research performed in 2021 or research study the authors describe in the article currently submitted?
What data should be considered to be novel? Results that were published in 2021 or the findings of research study reported in the article currently submitted to the journal?
Of course, difference can be found in terms of lifestyle and quality of life among patients (34001156), and the diagnostic tools used for the diagnosis of MHE may vary, but it is generally accepted that sleep disturbance regardless of the place of residence is one of the early signs of hepatic encephalopathy.(13193045, Lancet, 1954 Sep 4; 267(6836):454-7).
My comments here are concerned solely with the idea that the authors should highlight the novelty of their research. I cannot approve the manuscript for publication in this form.
Author Response
"Please see the attachment."

Reviewer 3 Report
Idea and aim of the research:
The aim is not clear.
Is the aim is to study sleep features in patients with CLD (line 30), or is it to study the cognitive function in patients with CLD, or the relation between them (as in the title). If it is the latter, they are both present in CLD and classified among the features of MHE. If the previous; the title should be: "the prevalence of either in CLD patients"????
Design & Methdology:
This is not a prospective "cross sectional" study (line 18, 76); this is a prospective cohort study.
No mention of patients' consents.
Exclusion criteria are not clear at all. The sentence starting from line 85 to line 89 is not clear at all. Are those patients included or not and why? the studied cohort should be homogenous to draw a conclusion.
There were 17 (out of 74) patients did not complete the tests and questionnaires (dropout as mentioned in the abstract and in line 149). So, the net result of the patient studied is 57 patients. Why statistics, tables, results and discussion are talking about 74 (not 57) patients. This is a clear flaw in the results, and so all the work.
Line 180 mentioned that comorbidities such as Diabetes Mellitus are equally distributed between all patients. Well, the results presented in Tab 3 do not tell so. (the part about Diabetes). Also the sentence from line 296 to 297 mentioned that diabetes mellitus was among the exclusion criteria.!!
All through the manuscript there is a mention of the term (steatosis/hepatitis) as in line 183 and 184. Does this mean NASH or NAFLD or both, this is not the usual term used.
Is there any explanation from the results of the sentence in line 336. Did the work involve follow up of patients after treatment. Also this contradicts the paragraph from line 264 to line 268.
In the Conclusion:
Line 353 mentioned something that was not previously present in the whole manuscript: "early stages of CLD". This was not written or explained before. The work included patients with decompensated liver cirrhosis Child C? Revise Table 1
some terms are not usual in scientific manuscripts:
What is the term morbimortality (line 38). Should be morbidity and mortality.
The word "Impossible" (line 57) should not be used in scientific writing. Should be difficult. The same with "done by convenience" (line 78).
Author Response
"Please see the attachment."

Round 2
Reviewer 1 Report
No further comments. The manuscript has been improved significantly and I approve of publication. Congratulations of this work.
Author Response
Thank you for taking time to review our paper and for your valuable suggestions that improved our manuscript.
Reviewer 2 Report
In light of the corrections and new information about the novelty the authors have introduced into the article, I am pleased to approve this article for publication.
Author Response

(The authors gave the same response as above.)

Reviewer 3 Report
Thank you for the effort to correct many of the highlighted comments. However:
Title: what is meant by the word "associations"?
The aim/idea: does not add much to the already known and written in the literature; that sleep disturbances and cognitive impairment are among the early manifestations of CLD associated with portal hypertension i.e. MHE.
Through the manuscript: The term "pre-cirrhosis" is not scientific. The term "CLD" is the usual term (which was used in some parts of the manuscript!!), authors should state what do they mean by the terms: CLD and pre-cirrhosis, with references. Recently the term cACLD have been proposed (kindly revise BAVENO VII recommendations 2022, which is an important guide in practice of portal hypertension and its sequalae, and should also be included in the references.)
In the inclusion/exclusion criteria; authors should mention the classification/types of HE. Then state which type is the focus of the research. Patients with acute hepatitis (regardless of the cause) should also be excluded. To exclude type A HE.
Also, exclusion criteria stated that patients with sig. alcohol intake were excluded. However; in line 178: " Overall, we found that the predominant etiology of CLD was related to alcohol consumption". !
In the paragraph from line 274 to line 281: it is well known (in the literature) that sleep disturbances are one of the manifestations of MHE. So, what is idea in this paragraph.?
Was there a specialist hepatologist among the authors?
